# Prolactin-mediates a lactation-induced suppression of arcuate kisspeptin neuronal activity necessary for lactational infertility in mice

Eleni CR Hackwell[1,2], Sharon R Ladyman[1,2,3], Jenny Clarkson[1,4], H James McQuillan[1,2], Ulrich Boehm[5], Allan E Herbison[6], Rosemary SE Brown[1,4], David R Grattan[1,2,3]*

[1]Centre for Neuroendocrinology, School of Biomedical Sciences, University of Otago, Dunedin, New Zealand; [2]Department of Anatomy, School of Biomedical Sciences, University of Otago, Dunedin, New Zealand; [3]Maurice Wilkins Centre for Molecular Biodiscovery, Auckland, New Zealand; [4]Department of Physiology, School of Biomedical Sciences, University of Otago, Dunedin, New Zealand; [5]Saarland University School of Medicine, Centre for Molecular Signalling (PZMS), Experimental Pharmacology, Homburg, Germany; [6]Department of Physiology, Development and Neuroscience, University of Cambridge, Cambridge, United Kingdom

*For correspondence: dave.grattan@otago.ac.nz

Competing interest: The authors declare that no competing interests exist.

## eLife Assessment

This **fundamental** work advances our understanding of the mechanisms underlying lactation-induced infertility. **Compelling** evidence supports the notion that prolactin inhibits kisspeptin activity and LH pulsatile release and that loss of this signal results in an early reestablishment of fertility during lactation. This work will be of interest to endocrinologists and reproductive biologists.

**Abstract** The specific role that prolactin plays in lactational infertility, as distinct from other suckling or metabolic cues, remains unresolved. Here, deletion of the prolactin receptor (Prlr) from forebrain neurons or arcuate kisspeptin neurons resulted in failure to maintain normal lactation-induced suppression of estrous cycles. Kisspeptin immunoreactivity and pulsatile LH secretion were increased in these mice, even in the presence of ongoing suckling stimulation and lactation. GCaMP fibre photometry of arcuate kisspeptin neurons revealed that the normal episodic activity of these neurons is rapidly suppressed in pregnancy and this was maintained throughout early lactation. Deletion of Prlr from arcuate kisspeptin neurons resulted in early reactivation of episodic activity of kisspeptin neurons prior to a premature return of reproductive cycles in early lactation. These observations show dynamic variation in arcuate kisspeptin neuronal activity associated with the hormonal changes of pregnancy and lactation, and provide direct evidence that prolactin action on arcuate kisspeptin neurons is necessary for suppressing fertility during lactation in mice.

## Introduction

In mammals, lactation is accompanied by a period of infertility. This adaptive change establishes appropriate birth spacing to enable maternal metabolic resources to be directed towards caring for the newborn offspring, rather than supporting another pregnancy (*Short, 1976*). Lactational infertility

is characterised by a lactation-induced suppression of pulsatile luteinizing hormone (LH) secretion, and the temporary loss of the reproductive cycle (in rodents this is exhibited as an extended period of diestrus or anestrus) (*McNeilly et al., 1983*; *Fox and Smith, 1984*; *McNeilly, 2001*; *Tsukamura and Maeda, 2001*). Lactation is also characterised by chronically elevated levels of the anterior pituitary hormone prolactin, which is essential for milk production and promotes adaptive changes in maternal physiology and behaviour (*Short, 1976*; *McNeilly et al., 1983*; *McNeilly, 2001*; *Tsukamura and Maeda, 2001*). Despite hyperprolactinaemia being a well-recognised cause of infertility, the specific role that prolactin plays in lactational infertility, as distinct from other suckling- or metabolic-related cues, is currently unclear (*McNeilly, 2001*; *McNeilly, 1997*).

Recent in vivo studies have confirmed that kisspeptin neurons in the arcuate nucleus of the hypothalamus are responsible for the periodic release of gonadotrophin-releasing hormone (GnRH) and subsequent pulsatile LH secretion that drives reproductive function (*Clarkson et al., 2017*; *Han et al., 2019*; *McQuillan et al., 2019*; *Moore et al., 2022*; *Han et al., 2023*). Studies using GCaMP fibre photometry in conscious mice have demonstrated that the arcuate kisspeptin neuronal population exhibits episodes of increased intracellular calcium levels coincident with, and immediately preceding, each pulse of LH secretion in intact and gonadectomised male and female mice (*Clarkson et al., 2017*; *Han et al., 2019*; *McQuillan et al., 2019*). Miniscope investigations showed that individual kisspeptin neurons within the arcuate population act in a coordinated, synchronised, and episodic manner (*Moore et al., 2022*; *Han et al., 2023*). Loss of pulsatile LH secretion during lactation and consequent lactational infertility may be caused by the loss of kisspeptin-mediated stimulation of GnRH secretion (*Yamada et al., 2007*; *Liu et al., 2014*; *Roa et al., 2006*; *Ladyman and Woodside, 2014*; *True et al., 2011*; *Brown et al., 2014*; *Xu et al., 2009*). Kisspeptin expression is markedly suppressed in lactation (*Yamada et al., 2007*; *True et al., 2011*) and even when exogenously stimulated, kisspeptin neurons are unable to activate GnRH neurons during lactation, likely due to a lack of kisspeptin synthesis (*Liu et al., 2014*).

It is well established that hyperprolactinemia causes infertility, and thus, the elevated prolactin present in lactation seems a likely candidate to be involved in suppressing fertility during lactation. Prolactin administration acutely suppresses LH secretion (*Brown et al., 2019*), and chronic exposure to elevated prolactin reduces *Kiss1* mRNA expression in the arcuate nucleus (*Brown et al., 2014*; *Araujo-Lopes et al., 2014*; *Sonigo et al., 2012*). In lactating mice, suppressing endogenous prolactin secretion shortens the period of infertility (*Hackwell et al., 2023*), suggesting that prolactin is important for maintaining the suppression of pulsatile LH secretion during lactation. Such a role for prolactin is controversial, however (*McNeilly, 2001*; *McNeilly, 1997*; *McNeilly, 1994*; *Sugimoto et al., 2022*; *Tay et al., 1996*; *Díaz et al., 1989*; *Ördög et al., 1998*), with studies in a number of species suggesting that the neural stimulation of suckling may be more important than prolactin in maintaining lactational infertility (*Maeda et al., 1990*; *Smith, 1978*). However, it has previously been difficult to disentangle the specific role of prolactin, as suckling, prolactin, and milk production are so tightly linked that manipulating one ultimately impacts the others, making it difficult to determine the contribution of any one element. Here, using a conditional deletion strategy, we have blocked prolactin action in the brain leaving suckling, lactation, and maternal behaviour intact. Using GCaMP fibre photometry techniques, we have also documented arcuate kisspeptin neuron activity across pregnancy and lactation transitions in the same mice and established that prolactin directly acts on these neurons to suppress fertility in lactation.

## Results

### Prolactin action on forebrain neurons is necessary to maintain lactational infertility

Lactation has previously been associated with a marked decrease in *Kiss1* mRNA levels in both rostral periventricular regions of the third ventricle (RP3V) and arcuate nucleus populations (*Liu et al., 2014*). To determine whether prolactin was involved in the maintenance of lactational anestrus, the *Prlr* gene was knocked out of *Camk2a*-expressing neurons (most forebrain neurons) of female mice (*Prlr*^lox/lox/ *Camk2a*^Cre). Control *Prlr*^lox/lox mice showed a marked reduction in kisspeptin cell body immunoreactivity in the RP3V of lactating compared to intact, diestrous, nulligravid, non-lactating (henceforth termed NL) mice (p=0.0100, *Post hoc* Tukey's multiple comparisons test, *Figure 1A and C*; source data is

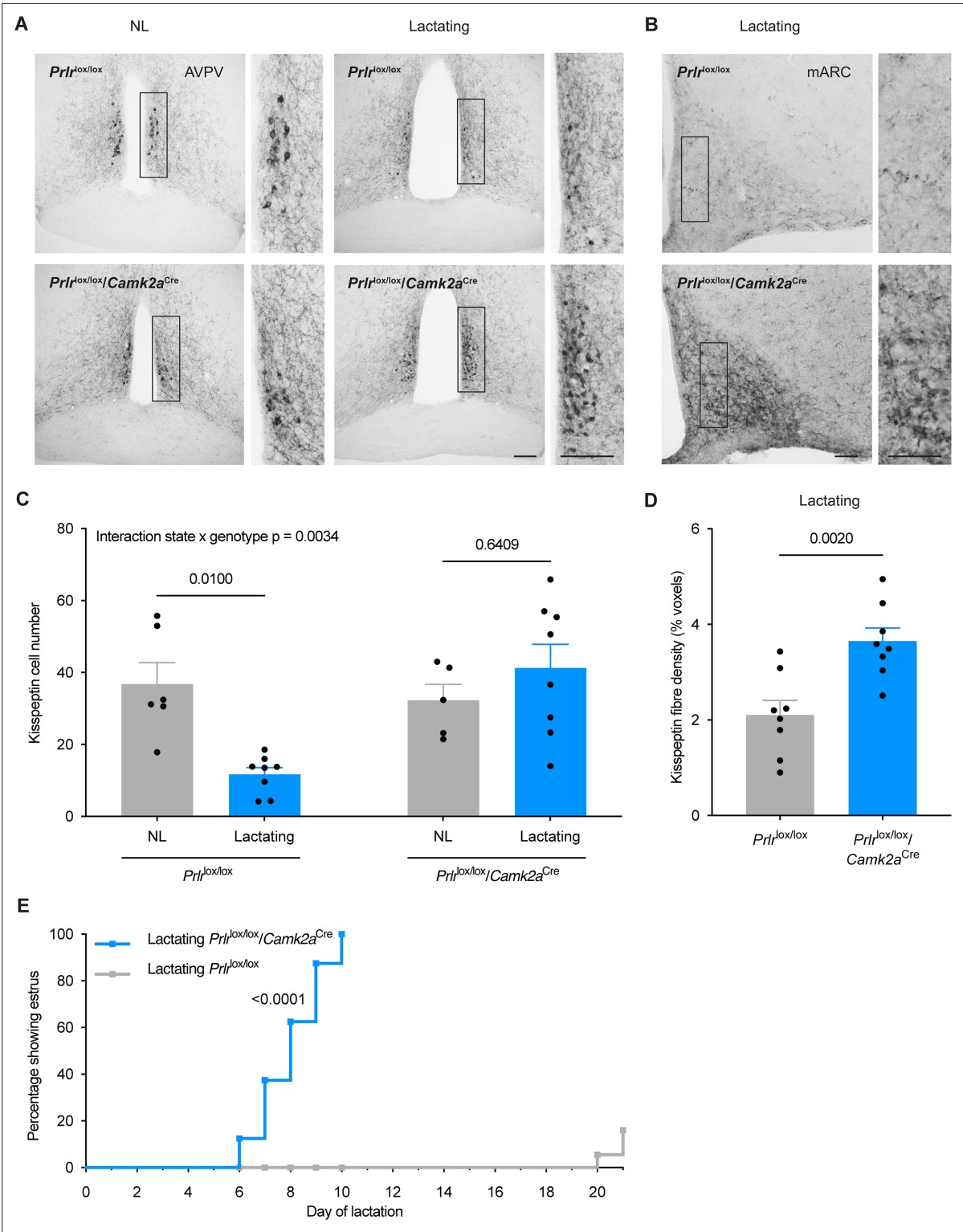

**Figure 1.** *Prlr*^lox/lox^/*Camk2a*^Cre^ mice do not undergo the normal period of lactational infertility and the lactation-induced suppression of kisspeptin immunoreactivity is absent. (**A**) Kisspeptin immunoreactivity is shown in representative photomicrographs from diestrus, nulligravid, non-lactating (NL; left), and lactating (right) *Prlr*^lox/lox^ control and *Prlr*^lox/lox^/*Camk2a*^Cre^ mice (from anteroventral periventricular nucleus (AVPV) region of RP3V). (**B**) Representative photomicrographs showing mid-arcuate nucleus (mARC) of lactating *Prlr*^lox/lox^ (top) and lactating *Prlr*^lox/lox^/*Camk2a*^Cre^ mice (bottom). (**C**) Total kisspeptin cell number for the RP3V (NL *Prlr*^lox/lox^ (n=6) versus lactating *Prlr*^lox/lox^ control (n=8) p=0.0100, NL *Prlr*^lox/lox^/*Camk2a*^Cre^ (n=5) versus lactating *Prlr*^lox/lox^/*Camk2a*^Cre^ (n=8) p=0.6409). Two-way ANOVA followed by Tukey's multiple comparisons test. (**D**) Quantification of kisspeptin fibre

*Figure 1 continued on next page*

*Figure 1 continued*

density in the arcuate nucleus (Fiji software, measured in percentage voxels per region of interest), showing total kisspeptin fibre density in the arcuate nucleus (lactating $Prlr^{lox/lox}$ control n=8, lactating $Prlr^{lox/lox}$/$Camk2a^{Cre}$ n=7, p=0.0020, unpaired two-tailed t-test). (**E**) Lactating $Prlr^{lox/lox}$/$Camk2a^{Cre}$ mice (blue, n=8) resume estrous cycles significantly earlier (100% within 6–10 d of lactation) than lactating $Prlr^{lox/lox}$ controls (grey, n=10) (p≤0.0001, Log-rank (Mantel-Cox) test). Scale bar image and insert = 50 µm. Values are shown as mean ± SEM.

The online version of this article includes the following source data and figure supplement(s) for figure 1:

**Source data 1.** Data for each of the graphs in *Figure 1*.

**Figure supplement 1.** Proportion of kisspeptin neurons showing prolactin receptor (*Prlr*) deletion using RNAscope in $Prlr^{lox/lox}$/$Camk2a^{Cre}$ mice.

**Figure supplement 2.** Gestational and maternal phenotyping of $Prlr^{lox/lox}$/$Camk2a^{Cre}$ mice and respective $Prlr^{lox/lox}$ controls.

available in supplementary file: *Figure 1—source data 1*). In contrast, in $Prlr^{lox/lox}$/$Camk2a^{Cre}$ mice the lactation-induced suppression of kisspeptin cell bodies in the RP3V was absent (p=0.6409, *Post hoc* Tukey's multiple comparisons test; interaction between reproductive state and genotype p=0.0034, two-way ANOVA, *Figure 1A and C*;). Fibre density in the arcuate nucleus was significantly increased in lactating $Prlr^{lox/lox}$/$Camk2a^{Cre}$ mice compared to lactating $Prlr^{lox/lox}$ controls (p=0.0020, unpaired two-tailed t-test, measured as percentage voxels within the region of interest, *Figure 1B and D*).

Estrous cycles during lactation were significantly altered by deletion of Prlr in the forebrain, with all $Prlr^{lox/lox}$/$Camk2a^{Cre}$ mice showing a return to estrus between day 6 and day 10 of lactation (*Figure 1E*), while, as normal, estrus did not occur until day 20 (after weaning) in control $Prlr^{lox/lox}$ mice (p≤0.0001, log-rank (Mantel-Cox) test, *Figure 1E*). No differences in litter weight gain from day 3 to day 8 of lactation were observed in either group (p=0.3282, mixed analysis test, *Figure 1—figure supplement 2*), indicating that the suckling stimulus that mice received and lactation itself was maintained in the absence of *Prlr* expression in *Camk2a* expressing neurons. Collectively, these data show that prolactin action in the brain is absolutely required for the lactation-induced suppression of kisspeptin expression and to maintain lactational infertility in mice.

In a separate cohort of $Prlr^{lox/lox}$/$Camk2a^{Cre}$ mice, pulsatile LH secretion was monitored in early lactation, prior to the return of estrous cycles. To rule out a potential role for progesterone in suppressing fertility during lactation (*McQuillan et al., 2019*; *Goodman and Karsch, 1980*; *Skinner et al., 1998*), the progesterone receptor antagonist mifepristone (RU486), was administered to these mice in early lactation. Vehicle-treated $Prlr^{lox/lox}$ control mice showed the expected near complete absence of pulsatile LH secretion during lactation (*Figure 2A and B*; source data is available in supplementary file: *Figure 2—source data 1*). In contrast, nearly all $Prlr^{lox/lox}$/$Camk2a^{Cre}$ mice showed a lack of the normal lactation-induced suppression of pulsatile LH secretion demonstrated by a significant increase in frequency of LH pulses compared to controls (effect of genotype p=0.0024, two-way ANOVA with Tukey's multiple comparisons test, *Figure 2A and B*). There was no effect of mifepristone on the pattern of LH secretion, suggesting that progesterone action is not required for the suppression of LH secretion (LH pulse frequency (interaction genotype x treatment p=0.2807; treatment p=0.8558; *Figure 2B*); mean LH levels (interaction genotype x treatment p=0.8572; treatment p=0.8457; *Figure 2C*); two-way ANOVA with Tukey's multiple comparisons test). These data indicate that prolactin is the primary signal responsible for the suppression of pulsatile LH secretion during lactation in mice.

## Episodic activity of arcuate kisspeptin neurons is suppressed during pregnancy and most of lactation

To directly assess the role of prolactin in regulating kisspeptin neuron activity during lactation, GCaMP fibre photometry was undertaken to monitor the real-time activity of arcuate kisspeptin neurons in freely behaving mice. We first undertook a longitudinal assessment of changes in kisspeptin neuronal activity by tracking individual $Kiss1^{Cre}$ mice throughout pregnancy, lactation, and following weaning (*Figure 3*; Raw fibre photometry data is available through Dryad, see Methods section: Statistical analysis for details).

Initially, GCaMP fibre photometry recordings were collected in the NL diestrous state, both with and without serial blood sampling to measure LH concentrations. As can be seen in *Figure 4—source data 1* is available in supplementary file: *Figure 4—source data 1*, arcuate kisspeptin photometry recordings were characterised by discrete synchronised events (SEs) of elevated intracellular calcium

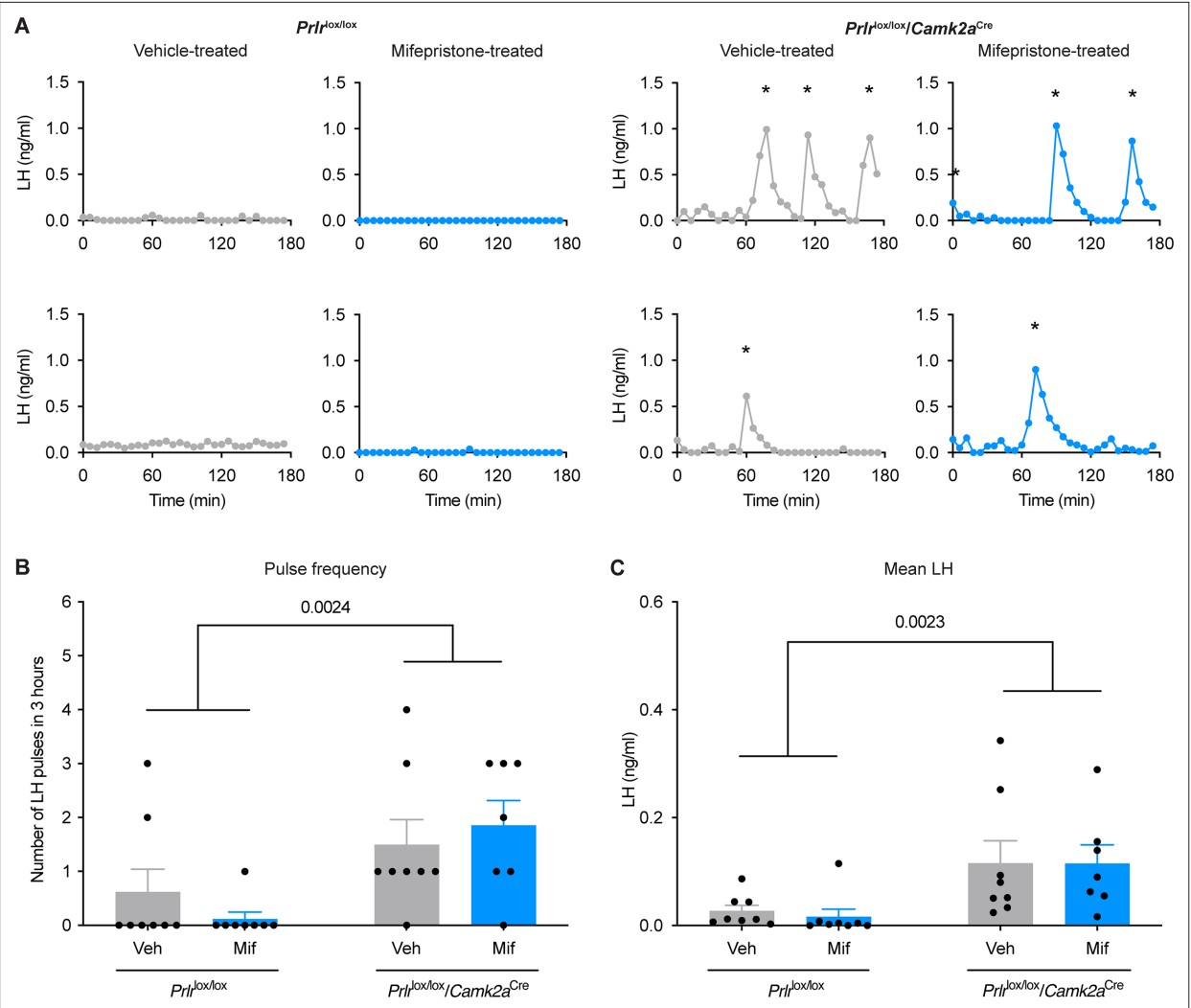

**Figure 2.** Prolactin action in the brain during lactation is necessary for the suppression of pulsatile luteinizing hormone (LH) secretion. Examples of pulsatile LH levels in the blood from lactating *Prlr*^lox/lox control and lactating *Prlr*^lox/lox/*Camk2a*^Cre mice that have either been treated with vehicle (sesame oil, s.c., grey, veh) or 4 mg/kg mifepristone (in sesame oil, s.c., blue, mif). Graphs show LH pulse frequency (B; interaction p=0.2807, genotype p=0.0024, state p=0.8558), and mean LH levels (C; interaction p=0.8572, genotype p=0.0023, state p=0.8457). Lactating vehicle-treated *Prlr*^lox/lox (n=8), lactating mifepristone-treated *Prlr*^lox/lox (n=8), lactating vehicle-treated *Prlr*^lox/lox/*Camk2a*^Cre (n=8), lactating mifepristone-treated *Prlr*^lox/lox/*Camk2a*^Cre (n=7). Asterisks indicate LH pulse peaks as detected by PULSAR Otago analysis. Two-way ANOVA followed by Tukey's multiple comparisons test. Values are shown as mean ± SEM.

The online version of this article includes the following source data and figure supplement(s) for figure 2:

**Source data 1.** Spreadsheet containing LH data from each individual sample.

**Figure supplement 1.** Pulsatile luteinizing hormone (LH) secretion profiles of *Prlr*^lox/lox/*Camk2a*^Cre mice and *Prlr*^lox/lox controls following vehicle or mifepristone treatment.

**Figure supplement 2.** Mifepristone dose pilot study and effect on litter weight gain.

(indicative of synchronous activity of the kisspeptin population), with each SE consistently correlating to a single pulse of LH secretion in the minutes following (p≤0.0001, chi-squared test). These were observed to be at a similar rate to that described previously in diestrous mice using GCaMP photometry in a different *Kiss1* mouse line (**McQuillan et al., 2019**), with periodic SEs occurring about once per hour (1.25±0.25 /hr; **Figures 3 and 4B**).

The activity of the arcuate kisspeptin population in *Kiss1*^Cre mice dynamically changed depending on the reproductive state of the mouse (p=0.0040, mixed-effect analysis, **Figures 3 and 4B**). On day 4 of pregnancy, SE frequency had markedly decreased (0.29±0.14 /hr; **Figures 3 and 4B**), indicating an

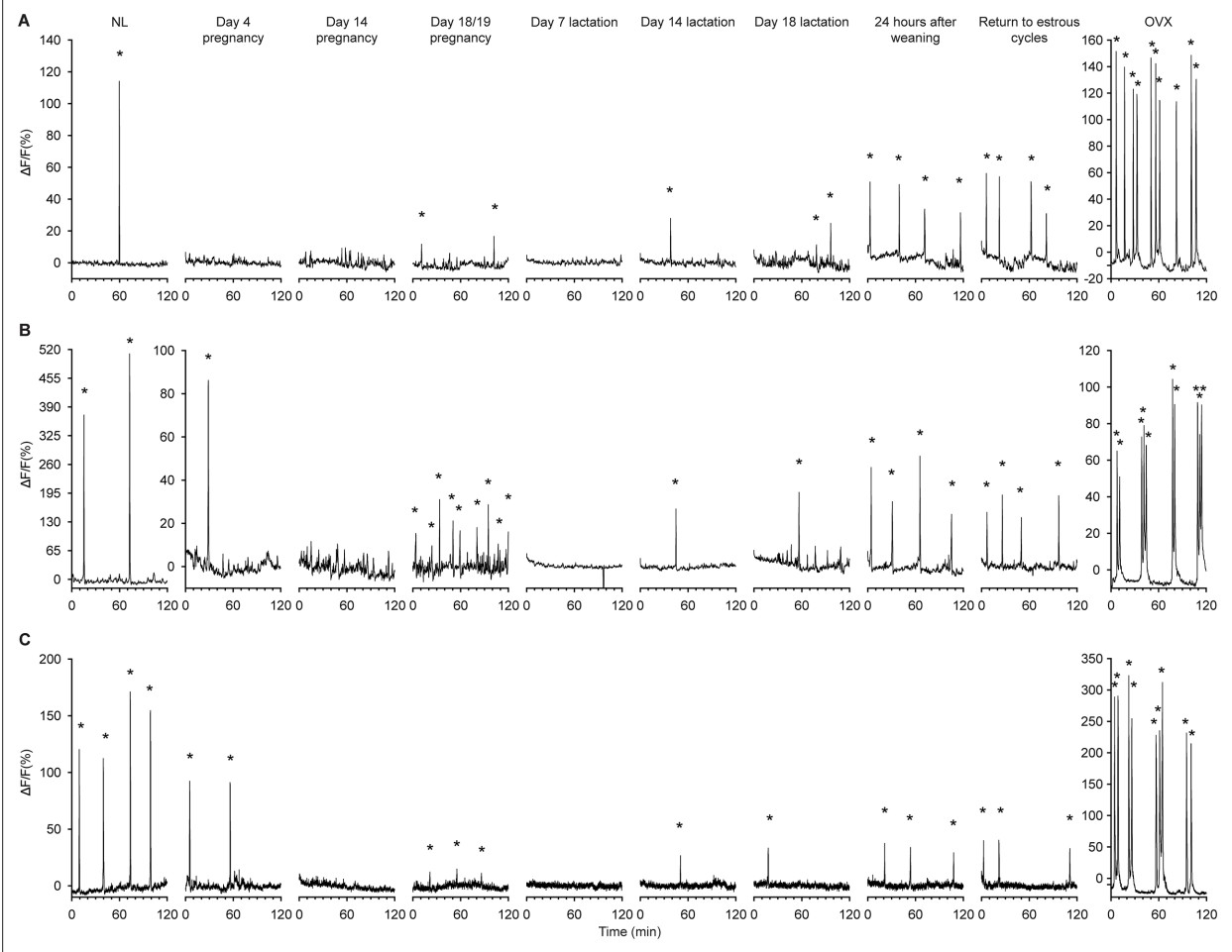

**Figure 3.** Longitudinal recordings of arcuate kisspeptin neuron GCaMP population activity throughout different reproductive states in the same *Kiss1*^Cre mice. Representative neuronal activity from three *Kiss1*^Cre mice throughout the NL (diestrus, nulligravid, non-lactating), pregnant, lactating, and post-weaning states. The time points monitored in order were: diestrus NL, day 4 pregnancy, day 14 pregnancy, day 18/19 pregnancy (overnight), day 7 lactation, day 14 lactation, day 18 lactation, 24 hr after weaning (day 22 postpartum), return to normal cycling following weaning (return to estrous cycles), and 10 d post ovariectomy (OVX). Asterisks indicate SEs. Note: change in y-axes scale on all three OVX timepoints and mouse shown in dataset (**B**) from day 4 of pregnancy onwards.

The online version of this article includes the following figure supplement(s) for figure 3:

**Figure supplement 1.** Miniature synchronised event (SE)-like activity on day 14 of pregnancy in *Kiss1*^Cre mice does not result in pulsatile luteinizing hormone (LH) secretion.

early reduction in activity of arcuate kisspeptin neurons during pregnancy. By day 14 of pregnancy, no SEs were seen (0±0 /hr; **Figures 3 and 4B**) and this was confirmed by a lack of pulsatile LH secretion (**Figure 3—figure supplement 1**). In late pregnancy (day 18), neuronal activity was monitored for 14 hr, during which time low amplitude, SEs were unexpectedly observed at similar frequencies to NL levels (2.18±1.20 /hr; **Figures 3 and 4B–C**). This unusual pattern of activity is illustrated in more detail in **Figure 5** where, alongside the resurgence of low amplitude SEs, a marked increase in baseline activity was observed relative to recordings at other reproductive stages (Raw fibre photometry data is available through Dryad, see Methods section: Statistical analysis for details; other source data is available in supplementary file: **Figure 5—source data 1**). This activity was reminiscent of the miniature SEs observed to be caused by activation of subgroups of arcuate kisspeptin cells using brain slice calcium imaging (**Han et al., 2023**), but we are unable to resolve such SEs using the present methods. Since our aim was to continue longitudinal assessment of the arcuate kisspeptin population into lactation and weaning, mice were not disrupted by blood sampling immediately prior to parturition as we were concerned that this additional stressor might interfere with the establishment of maternal

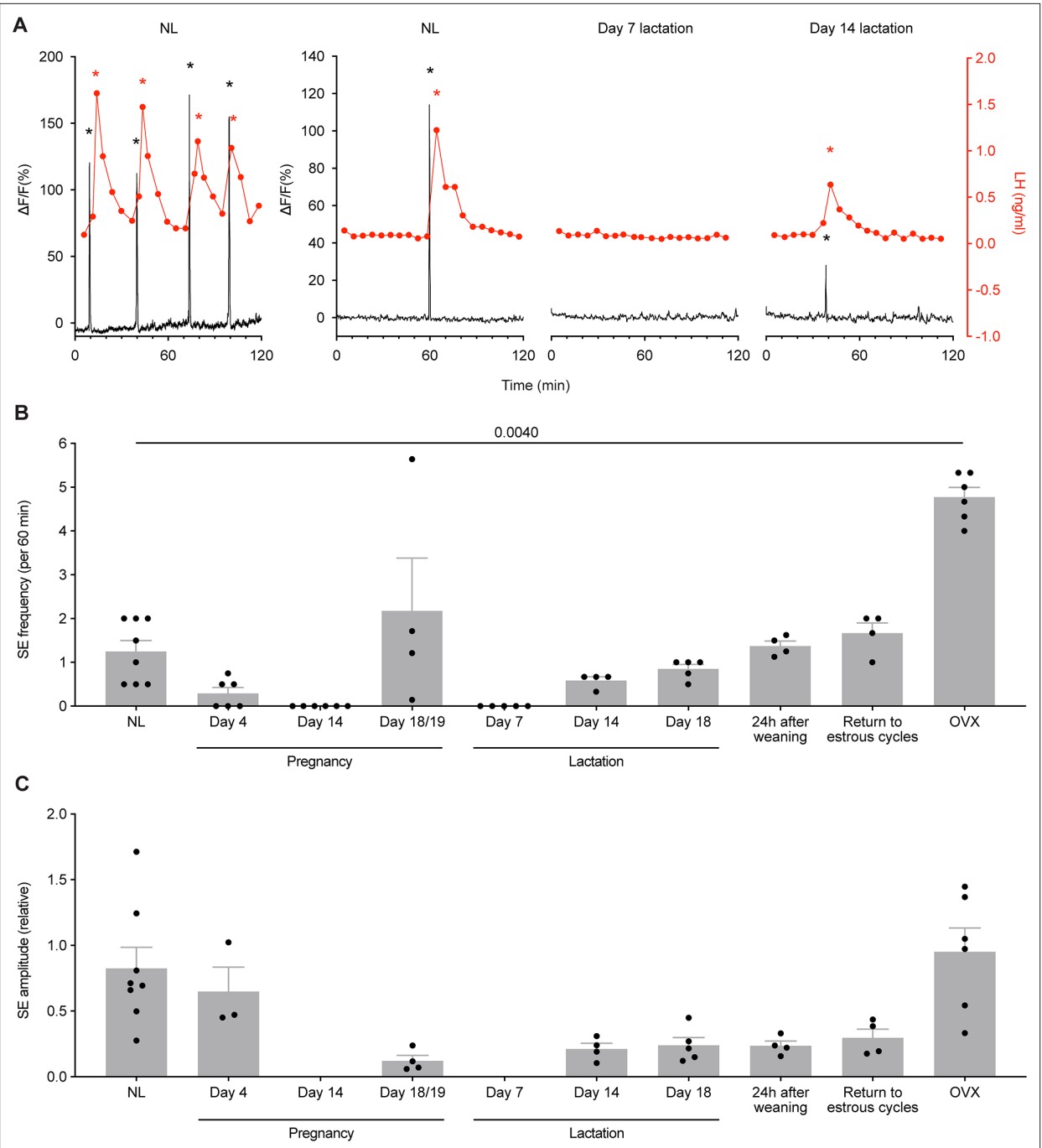

**Figure 4.** Synchronised Ca²⁺ events (SE) are consistently correlated with pulsatile luteinizing hormone (LH) secretion across different reproductive states in *Kiss1*^Cre mice. (**A**) When fibre photometry was paired with serial blood sampling for pulsatile LH secretion, the relationship between SEs and LH pulses was examined. Each time an SE was seen during a recording with blood sampling, pulsatile secretion of LH was also observed, with 100% correlation (p≤0.0001, chi-squared test; 73 out of 73 SEs observed led to an LH pulse). Representative examples of paired photometry and blood sampling are shown from the NL (diestrus, nulligravid, non-lactating) state, from day 7 lactation, and from day 14 lactation. (**B**) Quantitative analysis of SE frequency per hour across different reproductive states in *Kiss1*^Cre mice (p=0.0040, mixed effect analysis (fixed type III) with Tukey's multiple comparisons tests). (**C**) Relative SE amplitude of normalised ΔF/F across different reproductive states (unable to undertake statistical analysis). Black asterisks indicate SEs, and red asterisks indicate LH pulse peaks as detected by PULSAR Otago analysis. Values are shown as mean ± SEM.

The online version of this article includes the following source data for figure 4:

**Source data 1.** Data for LH pulse analysis in *Figure 4B and C*.

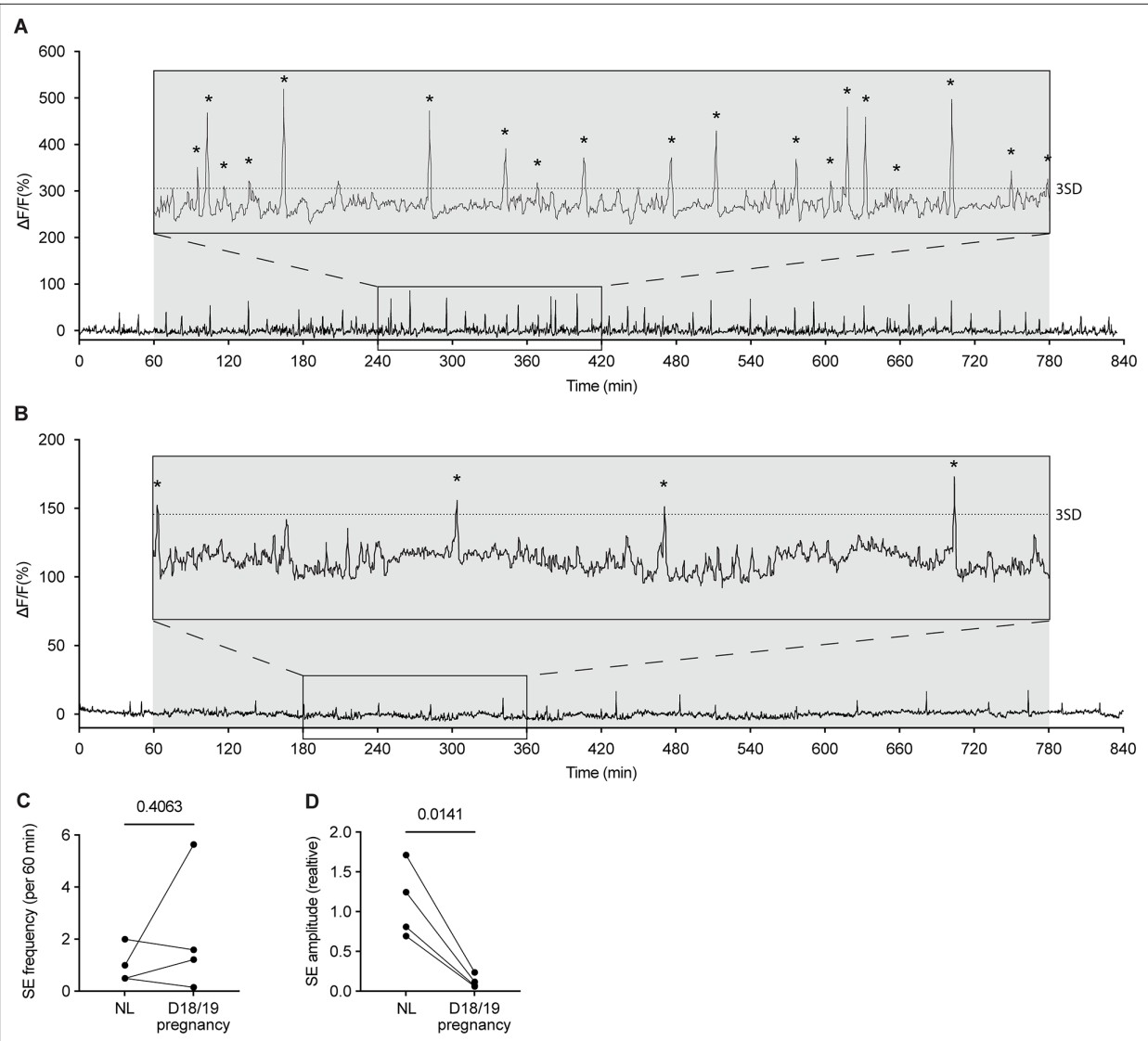

**Figure 5.** Activity of arcuate kisspeptin neurons on days 18 and 19 of pregnancy in *Kiss1*^Cre mice. Fibre photometry recordings of mice on the evening of day 18 of pregnancy (1800 hr) to the morning of day 19 of pregnancy (0800 hr) show low amplitude synchronised events (SEs). 3 hr section blown up for ease of viewing. (**C**) No difference was seen between the frequency of SEs (per 60 min) in the NL (diestrus, nulligravid, non-lactating) versus day 18/19 pregnancy (p=0.4063, paired two-tailed t-test), however, a significant decrease in relative SE amplitude is seen (D; p=0.0141, paired two-tailed t test). Asterisks indicate SEs. Dotted line in the insert of (**A**) and (**B**) indicates three standard deviations (3SD). Grey shaded region = lights off.

The online version of this article includes the following source data for figure 5:

**Source data 1.** Data for graphs *Figure 5C and D*.

behaviour. Hence, we are unable to report whether these low amplitude SEs and elevated baseline activity were associated with LH secretion.

Evaluation of activity of arcuate kisspeptin neurons during lactation showed complete suppression of activity on day 7 of lactation with a corresponding absence of pulsatile LH secretion (0±0 / hr; *Figures 3 and 4B*). This lactation-induced suppression of activity was partially relieved by day 14 of lactation (0.58±0.08 /hr; *Figures 3 and 4B*), with low frequency SEs again corresponding to low frequency pulsatile LH secretion. Further increases in SE frequency were seen on day 18 of lactation (0.85±0.10 /hr; *Figures 3 and 4B*), including an increase in baseline activity, similar to that seen in late pregnancy, and by 24 hr after weaning (day 22 postpartum) the frequency of SEs had returned to close to non-pregnant levels (1.38±0.11 /hr; *Figures 3 and 4B*). SE frequency remained unchanged on the day of the first diestrus following a return to estrous cycles after weaning (1.67±0.24 /hr; *Figures 3*

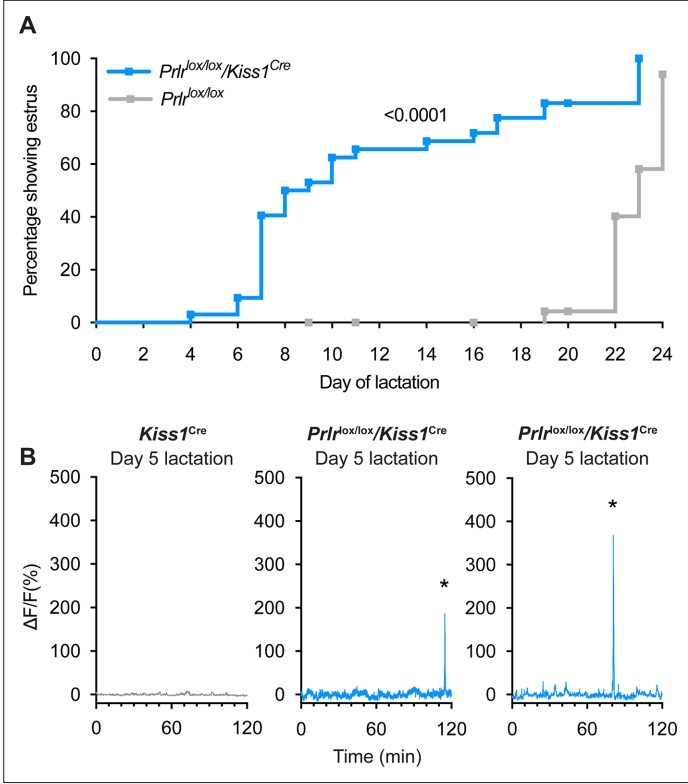

**Figure 6.** *Prlr*^lox/lox^/*Kiss1*^Cre^ mice do not undergo the normal period of lactational infertility and show early reactivation of arcuate kisspeptin neurons prior to estrus in lactation. (**A**) *Prlr*^lox/lox^/*Kiss1*^Cre^ mice resume estrous cycles significantly earlier (63% by day 10 of lactation, n=32, blue) than *Prlr*^lox/lox^ controls (4% by day 19 of lactation, n=30, grey) (p≤0.0001, Log-rank (Mantel-Cox) test). (**B**) Representative fibre photometry traces from day 5 of lactation from either a *Kiss1*^Cre^ control mouse or *Prlr*^lox/lox^/*Kiss1*^Cre^ mice. Mice with prolactin receptor (Prlr) knocked out of arcuate kisspeptin neurons (*Prlr*^lox/lox^/*Kiss1*^Cre^) show synchronised events (SEs) early in lactation. Asterisks indicate SEs.

The online version of this article includes the following source data and figure supplement(s) for figure 6:

**Source data 1.** Data for each of the graphs in *Figure 6*.

**Figure supplement 1.** Proportion of kisspeptin neurons showing prolactin receptor (*Prlr*) deletion using RNAscope in *Prlr*^lox/lox^/*Kiss1*^Cre^ mice.

**Figure supplement 2.** Gestational and maternal phenotyping of *Prlr*^lox/lox^/*Kiss1*^Cre^ mice and respective *Prlr*^lox/lox^ controls.

---

and 4B). Ten days post-ovariectomy (OVX), clusters of high amplitude arcuate kisspeptin population activity were observed (4.78±0.22 /hr; *Figures 3 and 4B*), consistent with previous reports following OVX in nulliparous mice (*McQuillan et al., 2022*). Collectively, these observations show extensive, dynamic variation in the activity of the arcuate kisspeptin neuronal population associated with pregnancy and lactation.

## Mice with an arcuate kisspeptin neuron-specific deletion have premature reactivation of estrous cycles and neuronal activity in lactation

To determine whether the prolactin-induced suppression of estrous cycles and pulsatile LH secretion was specifically mediated by kisspeptin neurons, mice were generated with an arcuate-specific deletion of the Prlr from kisspeptin neurons (*Prlr*^lox/lox^/*Kiss1*^Cre^) (*Brown et al., 2019*). Similar to the data from *Prlr*^lox/lox^/*Camk2a*^Cre^ mice, there was early resumption of estrous cycles in *Prlr*^lox/lox^/*Kiss1*^Cre^ mice during lactation (63% showing estrus by day 10 of lactation and 83% by day 19 lactation) compared to *Prlr*^lox/lox^ controls (4% by day 19 lactation) (p≤0.0001, log-rank (Mantel-Cox) test, *Figure 6A*; source

data is available in supplementary file: *Figure 6—source data 1*). No difference in litter weight gain during lactation was observed in either group indicating that suckling and/or lactation itself was not impaired (day 3–20 lactation weight gain, p=0.6404, two-way ANOVA, *Figure 6—figure supplement 2D*). In vivo, GCaMP fibre photometry in *Prlr*^lox/lox/*Kiss1*^Cre mice showed early reactivation of the arcuate kisspeptin population by day 5 of lactation (*Figure 6B*). This was accompanied by a clear return to estrus within this early lactation window in four out of five mice. These data demonstrate, in mice, that prolactin action specifically on arcuate kisspeptin neurons is responsible for maintaining suppression of those neurons, and thereby fertility, during lactation.

## Discussion

We demonstrate here that prolactin action in arcuate kisspeptin neurons is necessary for the maintained suppression of fertility during lactation in mice. Neuron-specific Prlr deletion (*Prlr*^lox/lox/*Camk-2a*^Cre) resulted in the premature return to estrus in early lactation, even in the presence of the ongoing suckling stimulus and the full metabolic consequences of milk production. Accompanying the resumption of estrus was an absence of the normal lactation-induced reduction in kisspeptin immunoreactivity (*Yamada et al., 2007*; *True et al., 2011*; *Araujo-Lopes et al., 2014*). Pulsatile LH secretion was also observed on day 5 of lactation prior to the premature estrus when it would normally have been completely absent (*Bohnet and Schneider, 1977*; *McNeilly, 1979*; *McNeilly et al., 1994*). To evaluate the specific role of kisspeptin neurons in mediating the prolactin-induced suppression of fertility, we have comprehensively mapped the activity of arcuate kisspeptin neurons throughout a full reproductive cycle: pregnancy, lactation, and after weaning in individual animals. The data show an immediate suppression of the activity of arcuate kisspeptin neuronal activity during pregnancy, and this is maintained throughout most of the lactation, apart from a brief window of reactivation immediately prior to parturition. Deleting Prlr specifically from arcuate kisspeptin neurons (*Prlr*^lox/lox/*Kiss1*^Cre) prevented the suppression of activity in early lactation, resulting in premature induction of episodic activation of kisspeptin neurons, and early onset of estrus. Combined, these data provide direct evidence that prolactin action on kisspeptin neurons is necessary for lactation-induced infertility in mice.

It is now well established that arcuate kisspeptin neurons form the GnRH 'pulse generator,' and hence drive the pulsatile release of GnRH from the hypothalamus and consequent pulsatile secretion of LH from the pituitary that is required for fertility (*Herbison, 2018*; *de Roux et al., 2003*; *Seminara et al., 2003*). This is the first study to monitor the activity of the arcuate kisspeptin neurons across different reproductive states in the same animal, and the data largely match previously described patterns of LH secretion (*McNeilly, 1979*; *Morishige et al., 1973*). The frequency and dynamics of the SEs changed dramatically, initially due to the pregnancy-induced changes in ovarian hormones. The abrupt decrease in arcuate kisspeptin activity in early pregnancy is likely caused by rising levels of progesterone, known to profoundly suppress the activity of arcuate kisspeptin neurons and LH secretion (*McQuillan et al., 2019*; *Goodman and Karsch, 1980*; *Skinner et al., 1998*). Progesterone is elevated throughout pregnancy, gradually increasing until luteolysis and progesterone withdrawal occur in the lead-up to parturition (*Virgo and Bellward, 1974*; *Pointis et al., 1981*; *Bridges, 1984*; *Sugimoto et al., 1997*). Interestingly, we observed a transient reactivation of the arcuate kisspeptin neurons in the night between days 18 and 19 of pregnancy. This was characterised by frequent, low amplitude SEs, and increased baseline activity that may represent the intermittent synchronised activity of small subsets of arcuate kisspeptin neurons that have not yet transitioned to full synchronisation of the whole population (*Han et al., 2023*). It seems likely that this pattern of activity is associated with progesterone withdrawal in late pregnancy and may be important in stimulating follicular growth leading up to a postpartum ovulation (*Thapa et al., 1988*; *Zakar and Hertelendy, 2007*).

In early lactation, the episodic activity of arcuate kisspeptin neurons was absent, with sporadic low-amplitude activity returning around day 14 of lactation. There was another period of increased baseline activity in late lactation, similar to that seen in late pregnancy, potentially representing a signature of reactivation of synchronised activity of the arcuate kisspeptin neurons. This increase in arcuate kisspeptin population activity during late lactation mirrors the increase in LH levels that has been reported as lactation progresses (*Rolland et al., 1975*). Overall patterns of SE activity rapidly returned to normal diestrus levels soon after weaning (*Rolland et al., 1975*). However, in the absence of Prlr in arcuate kisspeptin neurons, SE activity reappeared as early as day 5 of lactation, even in the presence of ongoing suckling. These data clearly show that prolactin action in arcuate kisspeptin neurons

is necessary to sustain lactational infertility in mice. The observed disruption of lactational infertility in $Prlr^{lox/lox}/Kiss1^{Cre}$ mice is particularly remarkable given that Prlr deletion is restricted to arcuate kisspeptin neurons in this model (**Brown et al., 2019**), and prolactin action on RP3V kisspeptin neurons (**Sonigo et al., 2012**; **Kokay et al., 2011**) and on gonadotrophs in the pituitary gland (**Hodson et al., 2010**; **Tortonese et al., 1998**; **Henderson et al., 2008**) are unaffected.

The indispensable role of prolactin in mediating lactation-induced infertility in the mouse is surprising, given the consensus of much work in other species concluding that other factors may be more important (see **McNeilly, 2001**; **McNeilly, 1997**; **Tay et al., 1996**; **Díaz et al., 1989**; **Woodside and Jans, 1995**). This may reflect a level of redundancy amongst contributing factors across all species, including ovarian hormones, metabolic cues, and neural inputs of suckling. Notably, the conditional deletion approach described here distinguishes prolactin action from the neurogenic effects of suckling without altering the process of lactation itself. Moreover, this approach avoids the potential confounding effects of using dopamine agonists to suppress prolactin (**Maeda et al., 1990**; **Smith, 1978**), given that dopamine can directly inhibit GnRH neuronal activity (**Liu and Herbison, 2013**).

While the effects of widespread neuronal deletion ($Prlr^{lox/lox}/Camk2a^{Cre}$) on fertility were largely recapitulated by the arcuate kisspeptin-specific model ($Prlr^{lox/lox}/Kiss1^{Cre}$), it was apparent that the global deletion was more effective at inducing the return to estrus during lactation (in 100% of $Prlr^{lox/lox}/Camk2a^{Cre}$ mice by day 10), compared to the arcuate kisspeptin-specific model (63% by day 10, and 83% by day 19). This may be due to the absence of lactation-induced suppression of $Kiss1$ expression in RP3V kisspeptin neurons of $Prlr^{lox/lox}/Camk2a^{Cre}$ mice. Similarly, we cannot rule out the possibility that other populations of prolactin-sensitive neurons, such as GABA or dopamine neurons (**Kokay et al., 2011**), may contribute to suppressing estrous cycles during lactation.

Nevertheless, our data collectively provide strong evidence that prolactin action on arcuate kisspeptin neurons is the primary factor mediating lactation-induced infertility in mice. Given that hyperprolactinemia induces infertility in many other mammalian species, including humans (**Sonigo et al., 2012**; **Hackwell et al., 2023**; **Greer et al., 1980**; **Evans et al., 1982**; **Koike et al., 1991**; **Patel and Bamigboye, 2007**; **Cohen-Becker et al., 1986**; **Sarkar and Yen, 1985**; **Fox et al., 1987**; **Park et al., 1993**; **Park and Selmanoff, 1991**; **Grattan et al., 2007**), it is possible that a conserved mechanism could be contributing to lactational infertility in these species.

## Materials and methods

### Animals

All experiments were performed using adult female mice on a C57BL/6 J background (RRID:IMSR_JAX:000664; 8–20 wk of age unless otherwise stated). Mice were housed under controlled temperature (22 °C±2 °C) and lighting (12 hr light/12 hr dark schedule, with lights on at 0600 hr) with ad libitum access to food and water (Teklad Global 18% Protein Rodent Diet 2918; Envigo, Huntingdon, United Kingdom). Daily body weight was recorded and daily vaginal cytology was used to monitor the estrous cycle stage. All experiments were carried out with approval from the University of Otago Animal Welfare and Ethics Committee.

To establish pregnancies, mice were mated with male wild-type C57BL/6 J mice (presence of sperm plug = day 1 pregnancy). Male mice were then removed once a sperm plug was seen, and no male mice were present at parturition. The first day a litter was seen was counted as day 1 of lactation and maternal mice were left undisturbed till day 3 of lactation, when vaginal monitoring would resume and litter size was normalised to 6 pups per animal, unless otherwise stated. The intention of the normalisation was to reduce variability in the degree of suckling stimulus each dam would receive.

In certain experiments, including towards the end of the fibre photometry experimental period, bilateral OVX was performed under isoflurane anaesthesia with pre-and post-operative Carprofen (5 mg/kg, s.c.) administered for pain relief. This enabled us to examine post-OVX changes in the activity of kisspeptin neurons and the expression of kisspeptin in the arcuate nucleus.

### Pulsatile hormone measurement

To monitor the pulsatile secretion of LH, serial tail tip blood sampling and measurement of LH by ELISA was undertaken as reported previously (**Brown et al., 2019**; **Steyn et al., 2013**; **Czieselsky**

*et al., 2016*). As novel exposure and restraint stress has been shown to suppress pulsatile LH secretion (*Briski and Sylvester, 1987*), all mice were habituated to the tail tip blood sampling procedure by picking up the mouse in a gentle restraint device (soft cardboard tube) or hand and lightly massaging the tail for approximately 5 min per day for at least 3 wk prior to experimentation (*Steyn et al., 2011*). Sequential whole blood samples (4 µl) were collected in 6- min intervals for 3 hr between 0900 and 1200 hr, unless otherwise stated. Samples were immediately diluted in 48 µl 0.01 M PBS/0.05% Tween 20, and frozen on dry ice before being stored at –20 °C for subsequent LH measurement.

## RNAscope

Mice were transcardially perfused with a micro-perfusion pump with 2% paraformaldehyde (PFA). Brain sections (14 µm-thick) were prepared, thaw-mounted onto superfrost-plus microscope slides, and then stored at –80 °C. RNAscope in-situ hybridisation was performed using the RNAscope 2.5 High definition Duplex Detection kit – chromogenic (Advanced Cell Diagnostics, Hayward, CA) largely in accordance with the manufacturer's instruction. The channel 1 Prlr probe was custom-designed to pick up only the long form of the prolactin receptor. It was designed to transcript NM_011169.5 with a target sequence spanning nucleotides 1107–2147 (Ref: 588621; Advanced Cell Diagnostics, Hayward, CA). The channel 2 Kiss1 probe was custom-designed to transcript NM_178260.3 with a target sequence spanning nucleotides 5–485 (Ref: 500141-C2; Advanced Cell Diagnostics, Hayward, CA). Sections were thawed at 55 °C, postfixed for 3 min in 2% PFA, washed in 0.01 M PBS for 5 min, and endogenous peroxidases were blocked with a hydrogen peroxidase solution for 10 min. Tissue was washed in distilled water (3x2 min), then briefly immersed in 100% ethanol, air dried for 5 min, and a hydrophobic barrier was applied. Tissue was permeabilised with RNAscope protease plus for 30 min a 40 °C. Sections were washed (2×2 min) and were hybridised with the Prlr and Kiss1 probes (1:300 dilution, Prlr:Kiss1) or negative control probe (Cat#320751; Advanced Cell Diagnostics, Hayward, CA) at 40 °C for 2 hr. Amplification (Amp 1–6) was performed in accordance with the manufacturer's instructions. Sections were then hybridised with a Fast-RED (1:60, Fast-RED B:Fast-RED A) for 10 min at room temperature, before undergoing further amplification steps (Amp 7–10) in accordance with the manufacturer's instructions. The final positive hybridisation was detected by incubation with the secondary detection reagents (1:50, Fast-GREEN B:Fast-GREEN) for 10 min at room temperature. Sections were washed, counterstained with haematoxylin (25% Gills), dried at 60 °C for 20 min, and cover-slipped with VectaMount (Vector laboratories, H-5000) before imaging using an Olympus BX51 light microscope and Olympus UPlanSApo 10/20 x lenses.

Quantification of the proportion of kisspeptin neurons co-expressing *Prlr* mRNA was undertaken in FIJI software (National Institute of Health, Bethesda, Maryland, USA) following image acquisition. Positive hybridisation for *Kiss1* and *Prlr* was determined based on scoring guidelines included in the manufacturer's protocol (Advanced Cell Diagnostics). The total number of *Kiss1*-expressing cells and the total number of these that showed *Prlr* mRNA expression were counted.

## Effect of neuron-specific deletion of the prolactin receptor gene on the maintenance of lactational infertility

To investigate whether prolactin action in the brain is required for lactational infertility, neuron-specific *Prlr* knockout mice (*Prlr*^lox/lox^/*Camk2a*^Cre^) and their respective Cre-negative controls (*Prlr*^lox/lox^) were generated (as previously described in *Brown et al., 2016*). We have previously shown that while *Prlr*^lox/lox^/*Camk2a*^Cre^ mice do not have a complete Prlr deletion in the forebrain. There are areas of extensive deletion (as measured by reduced prolactin-induced pSTAT5), such as the arcuate nucleus and ventro-medial nucleus of the hypothalamus, and there are also areas where Prlr is reduced by about 50% such as the medial pre-optic area (*Brown et al., 2016*; *Gustafson et al., 2020*). In our experience with these animals, this is sufficient to retain normal maternal behaviour in most animals (our unpublished data), and this was the case for the animals used in the present study. RNAscope in-situ hybridisation was done to confirm the degree of knockout in NL mice and 14- day post-OVX mice (*Prlr*^lox/lox^ intact n=6, *Prlr*^lox/lox^ OVX n=5, *Prlr*^lox/lox^/*Camk2a*^Cre^ intact n=5, *Prlr*^lox/lox^/*Camk2a*^Cre^ OVX n=5; all aged 8–16 wk). Both intact and OVX mice were included as kisspeptin cell bodies are only visible in the RP3V of intact mice and in the arcuate nucleus of OVX mice, due to estradiol regulation (*Smith et al., 2005*). *Prlr*^lox/lox^/*Camk2a*^Cre^ mice showed a significant decrease in the percentage of *Kiss1*-expressing cells

co-expressing *Prlr* compared to controls in both the RP3V (p≤0.0001) and arcuate nucleus (p=0.0009) (unpaired two-tailed t-tests, *Figure 1—figure supplement 1A–D*).

*Prlr*^lox/lox^/*Camk2a*^Cre^ mice are hyperprolactinaemic due to impaired negative feedback of prolactin on hypothalamic dopamine neurons (*Brown et al., 2016*) and, therefore, show disrupted estrous cycles (showing recurrent pseudopregnancy-like cycles with long periods of diestrus of approximately 14 d between estrous stages). However, these mice are able to become pregnant and have normal pregnancies. All *Prlr*^lox/lox^/*Camk2a*^Cre^ mice were given a 250 μl subcutaneous injection of bromocriptine (5 mg/kg, 5% ethanol/saline; Tocris Bioscience Cat#0427) prior to being mated. This treatment was designed to reinstate an estrous cycle in *Prlr*^lox/lox^/*Camk2a*^Cre^ mice. Bromocriptine is an agonist for the type 2 dopamine receptor and inhibits prolactin secretion from the pituitary gland (*Moult et al., 1982*), thereby terminating the pseudopregnancy-like state and bringing the mice into proestrus the following day. Following treatment, all mice were then housed with a stud male.

For *Prlr*^lox/lox^/*Camk2a*^Cre^ mice, estrous cycles were monitored from day 3 of lactation until the first day of diestrus following a day of estrus (proestrus and estrus had to be observed prior to transcardial perfusion on the first day of diestrus). Brains were collected following transcardial perfusion for assessment of kisspeptin immunoreactivity. For every lactating *Prlr*^lox/lox^/*Camk2a*^Cre^ mouse (n=8), the brain of a *Prlr*^lox/lox^ control mouse (n=8) of the equivalent day (±1) of lactation was also collected. A group of NL mice of both genotypes (n=5–6) was also perfused for immunohistochemistry on diestrus.

To evaluate pulsatile LH secretion in early lactation (prior to the return of estrous cycles) and to determine whether progesterone played any role in regulating pulsatile LH secretion in lactation, additional groups of lactating *Prlr*^lox/lox^/*Camk2a*^Cre^ and *Prlr*^lox/lox^ control mice were generated. These mice were treated with either the progesterone receptor antagonist, mifepristone (4 mg/kg in sesame oil, s.c.; AK Scientific Inc Cat#J10622), or vehicle (n=7–8 per group) on the morning of day 4 of lactation and day 5 of lactation. This dose was selected as it was found to be sufficient to cause termination of pregnancy in wild-type C57BL/6 J mice (p=0.0072, chi-squared test, *Figure 2—figure supplement 2A*; pilot study, n=6 both groups). Neither vehicle nor mifepristone treatment had an effect on litter weight gain (interaction of time x genotype & treatment p=0.5322, two-way repeated measures ANOVA, *Figure 2—figure supplement 2B*). Mice underwent blood sampling for LH for 3 hr on day 5 of lactation, 30 min after treatment.

## Measurement of LH concentrations

An established sandwich ELISA method was used to determine LH concentration in diluted whole blood samples collected from mice (*Steyn et al., 2013*; *Czieselsky et al., 2016*). Briefly, a 96-well high plate was incubated with bovine monoclonal antibody (LHβ518b7, 1:1000 in 1xPBS; Dr. L. Sibley, UC Davis, CA, USA, RRID:AB_2665514) for 16 hr at 4 °C. Following incubation of standards, controls and experimental samples for 2 hr, plates were incubated in rabbit polyclonal LH antibody (AFP240580Rb; 1:10,000; National Hormone and Pituitary Program, NIH; RRID:AB_2665533) for 90 min, followed by incubation with polyclonal goat anti-rabbit IgG/HRP antibody (1:1000; DAKO Cytomation; RRID:AB_2617138) for 90 min. Finally, plates were incubated in OPD (o-phenylenediamine capsules; Sigma-Aldrich Cat#P7288) for 30 min. A standard curve for the detection of LH concentration was generated using serial dilutions of mouse LH-reference preparation peptide (Dr. A.E. Parlow, National Hormone and Pituitary Program, NIH; AFP5306A). LH levels were read using a standard absorbance plate reader (SpectraMax ABS Plus; Molecular Devices) at 490 nm and 630 nm wavelengths. The assay had a sensitivity of 0.04 ng/ml to 4 ng/ml, with an intra-assay coefficient of variation of 4.40% and an inter-assay coefficient of variation of 8.29%.

PULSAR Otago was used to define LH pulses (*Porteous et al., 2021*). Parameters used; Smoothing 0.7, Peak split 2.5, Level of detection 0.04, Amplitude distance 3, Assay variability 0, 2.5, 3.3, G(1)=3.5, G(2)=2.6, G(3)=1.9, G(4)=1.5, G(6)=1.2. Mean LH levels were calculated by averaging all LH levels collected during the experiment. Individual LH profiles from mice are shown in *Figure 2—figure supplement 1*.

## Assessment of kisspeptin expression
### Perfusion and fixation of tissue

Mice were anaesthetised with sodium pentobarbital (15 mg/mL, i.p.) and transcardially perfused with 4% PFA. Brains were removed, postfixed in the same solution, and cryoprotected overnight in 30%

sucrose before being frozen at –80 °C. Two sets of 30 µm thick coronal brain sections were cut using a sliding microtome, from Bregma 1.10 mm to –2.80 mm. Brain sections were kept in cryoprotectant solution (pH = 7.6) at –20 °C until immunohistochemistry was performed.

## Immunohistochemistry

Immunohistochemistry for kisspeptin in the RP3V and arcuate nucleus was performed as previously described (*Clarkson et al., 2009b*). Briefly, sections were incubated in polyclonal rabbit anti-kisspeptin primary antibody (AC 566, 1:10,000; a gift from A. Caraty, Institut National de la Recherche Agronomique, Paris, France; RRID:AB_2314709) for 48 hr at 4 °C. Sections were then incubated with biotinylated goat anti-rabbit IgG (1:200, Vector biolabs, Peterborough, GK; RRID:AB_2313606) for 90 min at room temperature, followed by incubation in an avidin-biotin complex (Elite vectastain ABC kit, Vector laboratories). The bound antibody-peroxidase complex was visualised using a nickel-enhanced diaminobenzidine (DAB) reaction, to form a black cytoplasmic precipitate.

Brain sections were imaged as described in the RNAscope section above. Quantification of kisspeptin neurons in the RP3V, was undertaken by manually counting all labelled neurons present in all three subdivisions: the anteroventral periventricular nucleus (AVPV); rostral preoptic periventricular nucleus (rPVpo); and caudal preoptic periventricular nucleus (cPVpo). Two sections per subdivision per mouse were averaged for each animal. As kisspeptin cell bodies in the arcuate nucleus were not easily observed, as previously reported (*Clarkson et al., 2009a*), kisspeptin fibre immunoreactivity was imaged using a Gryphax NAOS colour camera (Jenoptik) and evaluated using FIJI software and the voxel counter function (National Institutes of Health). Kisspeptin fibre density was measured in the arcuate nucleus across the three subdivisions; rostral arcuate (rARC), middle arcuate (mARC), and caudal arcuate (cARC). Two sections of each subdivision per animal were counted and then averaged across each animal to get a total number and reported as the total amount of voxels per ROI (voxel fraction).

## Characterisation of arcuate kisspeptin neuronal activity using GCaMP fibre photometry
### Stereotaxic surgery and AAV injections

Adult *Kiss1*$^{Cre}$ or *Prlr*$^{lox/lox}$/*Kiss1*$^{Cre}$ mice (2–3 months old) were anaesthetised with 2% Isoflurane, given local Lidocaine (4 mg/kg, s.c.) and Carprofen (5 mg/kg, s.c.) and placed in a stereotaxic apparatus. A custom-made unilateral Hamilton syringe apparatus holding one Hamilton syringe was used to perform unilateral injections into the arcuate nucleus. The needles were lowered into place (–0.14 mm A/P, +0.04 mm M/L, –0.56 mm DV) over 2 min and left in situ for 3 min before the injection was made. 1 µl AAV9-CAG-FLEX-GCaMP6s-WPRE-SV40 (1.3×10$^{-13}$ GC/ml, University of Pennsylvania Vector Core, Philadelphia, PA, USA) was injected into the arcuate nucleus at a rate of ~100 nl/min with the needles left in situ for 3 min prior to being withdrawn over a period of 6 min. This was followed by implantation of a unilateral indwelling optical fibre (400 µm diameter, 6.5 mm long, 0.48 numerical aperture (NA), Doric Lenses, Canada, product code: MFC_400/430–0.48_6.5 mm_SM3*_FLT) at the same coordinates. Carprofen (5 mg/kg body weight, s.c.) was administered for post-operative pain relief. After surgery, mice received daily handling and habituation to the fibre photometry recording conditions over 4–6 wk before experimentation began.

## GCaMP fibre photometry

Photometry was performed as reported previously (*McQuillan et al., 2019*). Fluorescence signals were acquired using a custom-built fibre photometry system made primarily from Doric components. Violet (405 nm) and blue (490 nm) fibre-coupled LEDs were sinusoidally modulated at 531 and 211 Hz, respectively, and focused into a 400 µm, 0.48 numerical aperture fibre optic patch cord connected to the mouse. Emitted fluorescence was collected by the same fibre and focused onto a femtowatt photoreceiver (2151, Newport). The two GCaMP6s emission signals were collected at 10 Hz in a scheduled 5 s on/15 s off mode by demodulating the 405 nm (non-calcium dependent) and 490 nm (calcium-dependent) signals. The power output at the tip of the fibre was set at 50 µW. Fluorescent signals were acquired using a custom software acquisition system (Tussock Innovation, Dunedin, New Zealand) and analysed using custom templates created by Dr. Joon Kim (University of Otago, Dunedin, New Zealand) based on mathematics and calculations similar to those previously described

(*Kim et al., 2016*; *Sherathiya et al., 2021*). Briefly, the fluorescent signal obtained after stimulation with 405 nm light was used to correct for movement artefacts as follows: first, the 405 nm signal was filtered using a savitzky-golay filter and fitted to the 490 nm signal using least linear square regression. The fitted 405 nm signal was then subtracted and divided from the 490 nm signal to obtain the movement and bleaching corrected signal. The output of these templates is 490-adjusted405/adjusted405, which was multiplied to get the final ΔF/F as a percentage increase. All photometry data is reported as ΔF/F(%).

All recordings were obtained from freely behaving mice between 0800 hr and 1200 hr (apart from 24 hr post-weaning recording (0900 hr to 1700 hr), and day 18/19 pregnancy recording (1800 hr to 0800 hr the following day)). SEs were defined as when ΔF/F exceeds 3 standard deviations (SD) above the trace mean. Manual SE shape analysis was performed in addition to the standard deviation method for certain datasets where necessary. SEs were counted manually to determine the frequency of SEs per 60 min. The between-animal variability in total signal means that changes in SE amplitude can only be reported as relative changes within an animal. Relative SE amplitude was calculated by using normalised ΔF/F data and then subtracting the peak of an SE from the nearest nadir to the rise of the SE and averaging that for the number of SEs in a recording. To obtain normalised ΔF/F, three pre-pregnancy datasets from each mouse were used to find the average maximum ΔF/F for that mouse. All datasets were then divided by that normalisation value to get normalised ΔF/F for each trace for each individual mouse. In these long-term longitudinal studies, some data points are missing due to issues with recordings (e.g. dirty fibre optic cannula or system battery issues), meaning that occasionally fewer mice are used for analysis at various time points. During experimentation, no mice lost cannulas, but a gradual reduction in the amplitude of the signal was observed over time.

Following experimentation, brains were collected in a similar way to that described above. Post-mortem analysis of cannula placement and GCaMP6 transfection in each animal were not done, because, with experience, the quality and characteristics of the SE recordings (including corresponding pulsatile LH secretion) in each animal was indicative of successful cannula placement and transfection of the GCaMP. In a preliminary the trial the green fluorescent protein (GFP) attached to the GCAMP was restricted to the caudal arcuate nucleus (n=3 mice, preliminary trial), with little evidence of spread into the rostral or middle arcuate. GFP was seen exclusively ipsilaterally, with no GFP observed on the contralateral side. Cannula placement in these trial animals was in the caudal arcuate.

## Monitoring the activity of arcuate kisspeptin neurons across different reproductive stages in the same mice

Adult *Kiss1*^Cre mice were 8–10 wk of age at the beginning of experiments, and up to 12 months old by the time of final recording (n=8 during pregnancy; n=6 during lactation, two mice were euthanised due to dystocia). Monitoring of vaginal cytology and weight was continuous from 1 wk pre-surgery till day 19 of pregnancy and resumed on day 3 of lactation until the end of the experiment. All handling stopped on day 19 of pregnancy to avoid potential compromise of parturition and onset of maternal behaviour.

To investigate the activity of the arcuate kisspeptin population across different reproductive states in the same animal, the following recording protocol was followed for all *Kiss1*^Cre mice, unless otherwise stated; diestrus NL, day 4 of pregnancy, day 14 of pregnancy, day 18/19 of pregnancy (overnight), day 7 of lactation, day 14 of lactation, day 18 of lactation, 24 hr after weaning, first diestrus after estrous cycles begin following weaning, and 10 d post-OVX. In addition, blood sample collection for paired LH measurement was done in diestrus NL state, on day 14 of pregnancy (in four mice, a maximum of six samples were collected around small 'peaks' in baseline), day 7 of lactation, and day 14 of lactation. During sampling, blood samples were taken at the usual 6- min intervals previously described, however, if a SE was observed and the next scheduled sampling was more than 3 min away, an extra blood sample was taken 2 min after the SE. Following this additional sampling, the next scheduled blood sample would be taken at its original time and then every 6 min after, till another SE was seen. Blood sampling was not carried out at additional time points as blood sampling was undertaken at least a week apart, and stress from repeated sampling was attempted to be kept at a minimum e.g., not blood sampling around the time of birth. Fibre photometry recordings were usually between 2–4 hr in length. The only longer recordings were undertaken on day 18/19 of pregnancy (14 hr) and 24 hr after weaning (8 hr). These

particular recording sessions were extended to determine whether there were any longer-term changes occurring in the activity of the arcuate kisspeptin population in the lead-up to parturition or following weaning of pups (states closely followed by postpartum estrus and resumption of normal estrous cycles, respectively).

### Effect of arcuate kisspeptin neuron-specific deletion of the prolactin receptor gene on the maintenance of lactational infertility and the activity of kisspeptin neurons during lactation

Kisspeptin-specific prolactin-receptor knockout mice ($Prlr^{lox/lox}$/$Kiss1^{Cre}$) and their respective Cre-negative controls ($Prlr^{lox/lox}$) were generated as previously described (*Brown et al., 2019*). $Prlr^{lox/lox}$/$Kiss1^{Cre}$ (n=27) and $Prlr^{lox/lox}$ control (n=30) dams underwent estrous cycle monitoring from day 3 of lactation onwards to determine whether mice showed an early resumption of estrous cycles. RNAscope in-situ hybridisation (as described above) was done to confirm knockdown in a subgroup of these mice, with $Prlr^{lox/lox}$/$Kiss1^{Cre}$ mice showing a significant decrease in the percentage of $Kiss1$-expressing cells co-expressing $Prlr$ compared to controls in the arcuate nucleus ($p \leq 0.0001$, unpaired two-tailed t-test, *Figure 6—figure supplement 1A–B*; $Prlr^{lox/lox}$ n=6, $Prlr^{lox/lox}$/$Kiss1^{Cre}$ n=7).

To determine whether the deletion of the prolactin receptor from arcuate kisspeptin neurons led to early reactivation of these neurons during lactation, adult $Prlr^{lox/lox}$/$Kiss1^{Cre}$ mice (n=5) and an additional $Kiss1^{Cre}$ control mouse (n=1) (8 wk old at the start of the experiment, and up to 14 mo at final recording timepoint) were set up for fibre photometry, as described above. Monitoring of vaginal cytology and weight was continuous from 1- wk pre-surgery till day 18 of pregnancy and resumed on day 3 of lactation. Recordings were undertaken in a similar timeline to that described above, however, no pregnancy recordings were done, and in early lactation recordings were performed every 2 d from day 3 to day 9 of lactation, before following the same protocol as described. No blood samples were taken over the lactation period in these mice. As described previously, recordings were kept between 2–4 hr, apart from the 24 hr after weaning recording (8 hr).

### Statistical analysis

Data are presented as mean ± SEM and all statistical analysis were performed with PRISM software 10 (GraphPad Software, San Diego, CA, USA) with a p-value of <0.05 considered as statistically significant. Individual symbols in graphs represent individual mice. Differences in kisspeptin cell number or fibre density were assessed using two-way ANOVAs with Tukey's multiple comparisons tests or t-tests, with both analyses using combined averages of each animal (averaged number of cells or fibre density across the three subdivisions of each nucleus to get the total number reported). Resumption of estrous cycles was analysed using Log-rank (Mantel-Cox) test chi-squared test. LH pulse frequency data and mean LH data were analysed using two-way ANOVAs with Tukey's multiple comparisons tests. SE frequency and amplitude throughout reproductive cycles were analysed using mixed effect analysis (fixed type III) with Tukey's multiple comparisons tests where appropriate and day 18/19 of pregnancy data were analysed using t-tests. Correlation between SE occurrence and LH pulses was assessed using the chi-squared test. All fibre photometry data used for quantitative analysis and comparison were from the first pregnancy and lactation. Raw fibre photometry data is available through Dryad (DOI https://doi.org/10.5061/dryad.8931zcs21) and all other data generated or analysed during the study are included in the source files available as supplements to the figures. A full list of probability values, inferential statistics, and degrees of freedom for all data can be found in *Supplementary file 1*.

### Acknowledgements

We would like to acknowledge the research assistance of Zin Khant-Aung, genotyping by Pene Knowles, and Dr. Joon Kim for assistance with the analysis of fibre photometry data. Financial support: This work was supported by the Health Research Council of New Zealand (grant number: 21–560) and the Lions Club of Dunedin South - administered by Perpetual Guardian (Otago Medical Research Foundation).

# Additional information

## Funding

| Funder | Grant reference number | Author |
| --- | --- | --- |
| Health Research Council of New Zealand | 21-560 | David R Grattan |

The funders had no role in study design, data collection and interpretation, or the decision to submit the work for publication.

## Author contributions

Eleni CR Hackwell, Conceptualization, Data curation, Formal analysis, Investigation, Visualization, Methodology, Writing – original draft, Project administration; Sharon R Ladyman, Conceptualization, Investigation, Visualization, Methodology, Writing – review and editing; Jenny Clarkson, H James McQuillan, Investigation, Methodology, Writing – review and editing; Ulrich Boehm, Resources, Writing – review and editing; Allan E Herbison, Conceptualization, Resources, Methodology, Writing – review and editing; Rosemary SE Brown, Conceptualization, Supervision, Visualization, Project administration, Writing – review and editing; David R Grattan, Conceptualization, Resources, Supervision, Funding acquisition, Visualization, Project administration, Writing – review and editing

## Author ORCIDs

Ulrich Boehm ⬛ https://orcid.org/0000-0003-2436-6907
Allan E Herbison ⬛ https://orcid.org/0000-0002-9615-3022
David R Grattan ⬛ https://orcid.org/0000-0001-5606-2559

## Ethics

All experiments were carried out with approval from the University of Otago Animal Ethics Committee (AUP 20-93) in accordance with the Animal Welfare Act of New Zealand.

Reviewer #1 (Public review): https://doi.org/10.7554/eLife.94570.3.sa1
Reviewer #2 (Public review): https://doi.org/10.7554/eLife.94570.3.sa2
Author response https://doi.org/10.7554/eLife.94570.3.sa3

# Additional files

## Supplementary files
MDAR checklist

Supplementary file 1. Table containing description of each statistical comparison.

## Data availability

All data generated or analysed during this study are included in the manuscript and supporting files.

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
