## [Editor Report · eLife Assessment]

This **fundamental** work advances our understanding of the mechanisms underlying lactation-induced infertility. **Compelling** evidence supports the notion that prolactin inhibits kisspeptin activity and LH pulsatile release and that loss of this signal results in an early reestablishment of fertility during lactation. This work will be of interest to endocrinologists and reproductive biologists.

---

## [Referee Report · Reviewer #1 (Public review)]

Summary:

In this paper, Hackwell and colleagues performed technically impressive, long-term, GCaMP fiber photometry recordings from Kiss1 neurons in the arcuate nucleus of mice during multiple reproductive states. The data show an immediate suppression of activity of arc Kiss1 neuronal activity during pregnancy that is maintained during lactation. In the absence of any apparent change in suckling stimulus or milk production, mice lacking prolactin receptors in arcuate Kiss1 neurons regained Kiss1 episodic activity and estrous cyclicity faster than control mice, demonstrating that direct prolactin action on Kiss1 neurons is at least partially responsible for suppressing fertility in this species. The effect of loss of prolactin receptors from CamK2a expressing neurons was even greater, indicating either that prolactin sensitivity in Kiss1 neurons of the RP3V contributes to lactational infertility or that other prolactin-sensitive neurons are involved. These data demonstrate the important role of prolactin in suppressing Kiss1 neuron activity and thereby fertility during the lactational period in the mouse.

Strengths:

This is the first study to monitor activity of the GnRH pulse-generating system across different reproductive states in the same animal. Another strength in the study design is that it isolated the effects of prolactin by maintaining normal lactation and suckling (assessed indirectly using pup growth curves). The study also offers insight into the phenomenon of postpartum ovulation in mice. The results showed a brief reactivation of arcuate Kiss1 activity immediately prior to parturition, attributed to falling progesterone levels at the end of pregnancy. This hypothesis will be of interest to the field and is likely to inspire testing in future studies. With the exceptions mentioned below, the conclusions of the paper are well supported by the data and the aims of the study were achieved. This paper is likely to raise the standard for technical expectations in the field and spark new interest in the direct impact of prolactin on Kiss1 neurons during lactation in other species.

Weaknesses:

A weakness in the approach is the use of genetic models that do not offer complete deletion of the prolactin receptor from targeted neuronal populations. A substantial proportion of Kiss1 neurons in both models retains the receptor. As a result, it is not clear whether the partial maintenance of cyclicity during lactation in the genetic models is due to incomplete deletion or to the involvement of other factors. In addition, results showing no impact of progesterone on LH secretion during lactation are surprising, given the effectiveness of progesterone-containing birth control in lactating women. While the authors assert their findings may reflect an important role for prolactin in lactational infertility in other mammalian species, that remains to be seen. Hyperprolactinemia is known to suppress GnRH release, but its importance in the suppression of cyclicity during the lactation is controversial. Indeed, in several species, the stimulus of suckling is considered to be the main driver of lactational fertility suppression. Data from rats shows that exogenous prolactin was unable to suppress LH release in dams deprived of their pups shortly after birth; both suckling and prolactin were necessary to suppress a post-ovariectomy rise in LH levels. The duration of amenorrhea does not correlate with average prolactin levels in humans, and suckling but not prolactin was required to suppress the postpartum rise in LH in the rhesus monkey. The protocol of this or other studies might result in discordant results; alternatively, mice may be an outlier in their mechanism of cycle suppression.

Comments on revised version:

I remain enthusiastic about this article, which has been substantially improved in this revision. However, I didn't feel the authors responded to any of the points I raised previously in my public review (see Weaknesses), for example by adding to the manuscript's discussion section. These are the larger, conceptual issues that speak to the value of the paper in the context of the existing literature. The authors could also state they feel they have addressed the issues raised sufficiently in the text.

---

## [Referee Report · Reviewer #2 (Public review)]

Summary:

The overall goal of Eleni et al. is to determine if the suppression of LH pulses during lactation is mediated by prolactin signaling at kisspeptin neurons. To address this, the authors used GCaMP fiber photometry and serial blood sampling to reveal that in vivo episodic arcuate kisspeptin neuron activity and LH pulses are suppressed throughout pregnancy and lactation. The authors further utilized knockout models to demonstrate that the loss of prolactin receptor signaling at kisspeptin cells prevents the suppression of kisspeptin cell activity and results in the early reestablishment of fertility during lactation. The work demonstrates exemplary design and technique, and the outcomes of these experiments are sophistically discussed.

Strengths:

This manuscript demonstrates exceptional skill with powerful techniques and reveals a key role for arcuate kisspeptin neurons in maintaining lactation-induced infertility in mice. In a difficult feat, the authors used fiber photometry to map the activity of arcuate kisspeptin cells into lactation and weaning without disrupting parturition, lactation, or maternal behavior. The authors used a knockout approach to identify if the inhibition of fertility by prolactin is mediated via direct signaling at arcuate kisspeptin cells. Although the model does not perfectly eliminate prolactin receptor expression in all kisspeptin neurons, results from the achieved knockdown support the conclusion that prolactin signaling at kisspeptin neurons is required to maintain lactational infertility. The methods are advanced and appropriate for the aims, the study is rigorously conducted, and the conclusions are thoughtfully discussed.

Comments on the latest version:

All comments and suggestions have been addressed by the authors in this revision.

---

## [Author Response]

The following is the authors’ response to the original reviews.

**Reviewer #1 (Recommendations For The Authors):**
I recommend being explicit regarding how the animals were habituated to blood sampling.

On lines 109-111 we have added a more detailed explanation of how mice were habituated to blood sampling. This includes details that mice were held and had their tails palpated for approximately 5 minutes per day.

Were any mice excluded due to loss or movement of the implant over time? Any details to allow replication of long-term measurements like this should be included.

No mice lost their cannulas during experimentation so we have added a sentence on this on lines 303 to 304 to this effect. We have also noted that there was a slight decrease in signal over the months of experimentation. A statement on line 318 has also been added that clarifies two mice lost between the pregnancy and lactation stages of experiment were euthanised due to dystocia.

The text states that synchronized episodic activity reappeared as early as 3 days after birth, citing Figure 6c as evidence. There is no 6c. Figure 6b shows day 5 after birth.

This has been corrected.

The methods state mRNA levels had to be "above background" to be counted as colocalization. At how many fold/what percent above background was a cell considered positive for expression?

Positive hybridisation was scored according to the manufacturer’s protocol and a statement to this effect has been added on line 144.

Please ensure figure titles or the data graphs explicitly give the genotype of the mice in all figures (or state the mice are wildtype).

Genotype has been added to figure titles where possible. Genotypes are always given in figure legends and tables and/or explicitly stated on the figure itself.

Figure 4's title states events are "perfectly" correlated, which is a subjective term. I recommend saying "consistently" or "temporally" correlated, depending on your meaning.

This has been amended to read “consistently correlated”

**Reviewer #2 (Recommendations For The Authors):**
The comments below aim to clarify the paper's methodology and results but do not detract from my overall enthusiasm for this work.- Given past studies demonstrating prolactin action in the brain, particularly the MPOA/MPN, is essential for maternal behavior, can the authors please clarify why this behavior is retained in the cam2a prlr knockout mice? The authors mention that prlr in the MPOA is only knocked down 50% compared to WT controls. Is this sufficient to retain maternal behavior?

In our experience 50% Prlr in the MPOA is sufficient to retain normal maternal behaviour in most animals including the ones in this experiment (our original paper describing this showed relatively normal behavior, for example, with a vGAT and vGlut-mediated knockouts, and even a double knockout – it was only when we achieved complete KO with an AAV-Cre that we saw failure of maternal behavior – Brown et al, PNAS 3;114(40):10779-10784 2017). We have added a statement on lines 157-159 regarding this. We have an additional paper in preparation specifically characterising the maternal behaviour and lactation outcomes in this line of mice, and we find most animals display normal maternal behaviour, with slightly impaired milk production in later lactation.

- Supplementary Figure 1. Can the authors please clarify the criteria for a cell to be positive for prlr? The methods state that the signal must be "above background level." How was the background measurement obtained? In the negative control?

As per above, scoring of positive hydribisation was done according to the manufacturer’s protocol and a statement to this effect has been added on line 144.

- Lines 310-314: This sentence describes RNAscope analysis of prlr knockdown in kisspeptin cells and refers to Extended Figure 3 - but I believe this is in Supplementary Figure 1.

This has been corrected.

- Figure 3-4: When mice return to estrous cycles, the amplitude of episodic kisspeptin neuron activity is the same as 24 hours after weaning, which appears much lower than in virgin females. Does this reach significance? If so, do the authors know why kisspeptin activity is still suppressed, and can they comment on why this may not affect estrous cyclicity?

This does not reach significance – see Supplementary Table 1 (4C) for statistics. Therefore, no further analysis was done. This question would need to be examined with a follow up experiment. Given the 5s on, 15 s off scheduled mode of recording used here, amplitude was not an extremely accurate measure and amplitude has been reported as relative within each mouse. There is also an additional issue of a gradual reduction in amplitude of signal over time in these long-term experiments – although it is true that much larger signals were detected after ovariectomy at the end of the experiment. At present, we have not tried to interpret whether the changes in amplitude are informative.

- Fiber photometry studies: Please indicate whether a post-mortem examination of GCaMP transfection and fiber photometry placement was conducted, and what region of the ARC was imaged.

Brains from these mice were collected, however postmortem analysis of cannula placement of GCaMP6 transfection was not carried out in all mice. This was based on our experience with this method, in that the quality and characteristic pattern of activity seen, as well as corresponding LH secretion following an SE, was indicative of successful cannula placement and transfection. Incorrectly placed cannular failed to show SEs. A trial was done with 3 mice and cannula placement was found to be in the caudal ARC (cARC) with GFP (attached to GCaMP) restricted to the cARC. A statement has been added on lines 306-313 regarding this.

- Were male mice removed before birth? Please add to the methods section if not included.

Yes, male mice were removed after a sperm plug was seen and were never present at parturition. We have inserted additional details on line 95 to this effect.

**Reviewer #3 (Recommendations For The Authors):**
(1) Line 172: n=7-8 per group, yet in Supplementary Figure 2, n=6 per group.

These are referring to different groups of mice. N=7-8 is referring to the group size of mice in Figure 2 that were given mifepristone or vehicle control. In contrast the Supplementary figure 2 n number refers to the mice in the pilot study. Additional n number for the pilot study has been added on line 194.

(2) Line 314: Extended = suppl; Figure 3 = 1.

This has been corrected.

(3) Line 451: Figure 6C, does not exist.

This has been corrected.

Line 590: Reference 23 could be replaced by Ordog T et al 1998 Am J Physiol 274,E665 because it is later and more relevant to the topic.

This reference has been replaced with the suggested reference.